# A Novel Nanoporous Adsorbent for Pesticides Obtained from Biogenic Calcium Carbonate Derived from Waste Crab Shells

**DOI:** 10.3390/nano13233042

**Published:** 2023-11-28

**Authors:** Fran Nekvapil, Adina Stegarescu, Ildiko Lung, Razvan Hirian, Dragoș Cosma, Erika Levei, Maria-Loredana Soran

**Affiliations:** 1Ioan Ursu Institute, Babeș-Bolyai University, 1 Kogălniceanu, 400084 Cluj-Napoca, Romania; fran.nekvapil@ubbcluj.ro (F.N.); razvan.hirian@ubbcluj.ro (R.H.); 2National Institute for Research and Development of Isotopic and Molecular Technologies, 67-103 Donat, 400293 Cluj-Napoca, Romania; adina.stegarescu@itim-cj.ro (A.S.); ildiko.lung@itim-cj.ro (I.L.); dragos.cosma@itim-cj.ro (D.C.); 3INCDO-INOE 2000, Research Institute for Analytical Instrumentation, 67 Donat, 400296 Cluj-Napoca, Romania; erika.levei@icia.ro

**Keywords:** crab shell, blue biotechnology, porosity, acetamiprid, adsorption

## Abstract

A novel nanoporous adsorbent was obtained through the thermal treatment and chemical wash of the wasted crab shells (BC1) and characterized by various techniques. The structure of BC1 at the end of the treatments comprised a mixture of calcite and amorphous CaCO_3_, as evidenced by X-ray diffraction and Fourier transform infrared absorption. The BET surface area, BET pore volume, and pore diameter were 250.33 m^2^ g^−1^, 0.4 cm^3^ g^−1^, and <70 nm, respectively. The point of zero charge of BC1 was determined to be around pH 9. The prepared adsorbent was tested for its adsorption efficacy towards the neonicotinoid pesticide acetamiprid. The influence of pH (2–10), temperature (20–45 °C), adsorbent dose (0.2–1.2 g L^−1^), contact time (5–60 min), and initial pesticide concentration (10–60 mg L^−1^) on the adsorption process of acetamiprid on BC1 was studied. The adsorption capacity of BC1 was 17.8 mg g^−1^ under optimum conditions (i.e., 20 mg L^−1^ initial acetamiprid concentration, pH 8, 1 g L^−1^ adsorbent dose, 25 °C, and 15 min contact time). The equilibrium data obtained from the adsorption experiment fitted well with the Langmuir isotherm model. We developed an effective nanoporous adsorbent for the recycling of crab shells which can be applied on site with minimal laboratory infrastructure according to local needs.

## 1. Introduction

Considerable effort was dedicated to develop various classes of nanomaterials including, but not limited to, carbons [1,2], metal–organic frameworks [3,4,5], and hierarchical nanoporous metals [6]. These nanomaterials hold potential for applications such as adsorbents, gas storage, photocatalysts, sensors, and many others. The common characteristics of these nanomaterials is a large specific surface area, but each of them has its particularities, described in more detail in the dedicated review papers cited above.

Referring to pore size, we may note three main classifications. According to the International Union of Pure and Applied Chemistry (IUPAC), pores are classified into (1) micropores, having a diameter less than 2 nm, (2) mesopores, between 2 and 50 nm, and (3) macropores, larger than 50 nm. However, here, we adopt the other approach which is used commonly in nanoscience, where a material with dimensions or features smaller than 100 nm can be considered a nanomaterial [7].

When talking about novel ultra-performant adsorbents, the artificial metal–organic frameworks (MOFs), covalent organic frameworks (COFs), synthesized porous carbon, and other similar advanced materials usually come to mind. Nature-derived materials are often overlooked, including crab shells. These are indeed one of the naturally formed minerals with various degrees of porosity. While not achieving impressive porosity properties compared to contemporary man-made materials, crab shells come from an interesting context—they represent a substantial proportion of inedible food waste, and their recycling into secondary products is also in line with the ever-increasing issue of proper waste management [8]. At this point, a deep scientific knowledge base is necessary to identify the appropriate type of shells with relatively high porosity compared to similar materials.

Since the porosity of the shell depends to a great extent on its biological origin, the selection of crab species, or at least the lower taxonomic group, is paramount. It dictates the porosity properties of the final nanomaterial. Here, we refer to the Atlantic blue crab (species *Callinectes sapidus*), which belongs to infraorder Brachyura (according to the World Register of Marine Species at https://www.marinespecies.org/, accessed on 2 October 2023). This group comprises decapod crustaceans with short tails curled under the wide thorax, which is quite rigid, with chitin fibers integrated into the magnesian calcite biomineral phase having an ordered pore canal system. According to the previous [9] and current thermogravimetry studies, the shell of crabs is composed of roughly 30% degradable organic matter (mostly chitin), while the rest of the mass represents the biomineral phase. Electron microscopy image analysis gives insight into pore canal dimensions, reported more precisely for several crab species. For instance, Yao et al. [10] reported the canal diameters of 40 to 50 nm in blue crab, 70 nm in stone crab, and 50 to 70 nm in dungeness crab. Nekvapil et al. [11] reported canal diameters of 42, 20, and 33 nm for the blue crab (*C. sapidus*) with white, blue, and red colored shells, respectively, and 45 nm for the green crab (*Carcinus aestuarii*) shell sourced in the southern Adriatic Sea. The geographical origin of source for crab shells might be important, as different oceanographic and habitat conditions may influence the shell biomineralization process; however, at this point, there is not enough published work for a conclusive discussion. The spacing between the pore canals is also regular, ranging from 34 to 190 nm, depending on the shell species of origin and type [10,11]. The length of these nanocanals presents a challenge, as it is difficult to obtain the longitudinal transect of the entire canal; however, the estimates are a few micrometers [10,11,12].

The pore surface area of these shells can be readily measured in native samples via nitrogen physisorption. It ranges from 7.17 to 11.05 m^2^ g^−1^ depending on the species [13,14,15]. This is in contrast with the fibrous and flexible exoskeletons of shrimps and lobsters, which, despite having a brittle and fibrous texture [12,16], are not suitable for BET measurement. Furthermore, they do not exhibit the kind of mineral phase porosity considered here. For this reason, the accuracy of porosity results reported in papers where a shell mixture of unspecified composition was used is questionable. The issue is that the measured value represents a mean of a vast range of materials with different porosity properties.

Recent studies have imaged in high resolution the ordered porosity of crab shells [10,11,12], and others have demonstrated that shells of the blue crab (*Callinectes sapidus*) have the capacity to be loaded with an anticancer drug 5-fluorouracil [13,17] and seaweed extract biofertilizer [18]. The respective substances are then released upon re-immersion in water. This testifies to the capacity of shell micropores and canals to house dissolved substances which remain within the bulk material after drying. Nekvapil et al. [15] have further shown that shell pore surface area and pore volume can be significantly increased through technically simple treatments, such as acetone immersion. It creates synergy between adsorbent material production and carotenoids extraction via a blue biotechnology approach within the overall context of works on the food waste management issue. Moreover, studies published in recent decades have shown that biogenic minerals, especially calcium carbonate crustacean and bivalve shells [19,20,21,22], indeed exhibit adsorption affinity towards certain pollutants.

The two major methods for converting waste crustacean shells to adsorbents are mineral biochar production [23,24,25] and simply rinsing and grinding native shells as they are [14,26,27,28]. Either method has its particularities, and the proper method must be applied in relation to the starting shell material composition. The mineral phase of the crab shells, after acid washes or thermal treatments, is usually considered in the context of heavy metal removal [29].

Acetamiprid is a neonicotinoid insecticide. It is used primarily and in its largest quantity for pest control in agriculture, but it is also used in horticulture, plant nurseries, and private turfs. Many formulations of agricultural products containing acetamiprid are available, such as liquids, wettable powder, wettable powder in soluble packets, soluble granules, etc. Although dietary, residential, aggregate, bystander, or occupational post-application exposure risks to humans are considered low, environmental risks seems to be specific for different kinds of biota. For instance, mammals, young plants, and aquatic invertebrates can experience acute and chronic intoxication upon consuming treated fruit or seeds, spray drifts, flooding, and runoffs, respectively. This pesticide does not seem to be directly toxic for fish, which rather act as bioaccumulation carriers [30].

The present work addresses and links the two global problems, food waste management and water contamination, through the concepts of a circular economy and knowledge-based blue bioeconomy. As a food waste item, blue crab shells were turned into a nanomaterial with significantly increased porosity relative to the starting material that is capable of adsorbing considerable quantities of the pesticide acetamiprid. These issues are also highlighted in major international policies, such as the UN’s Sustainable Development Goals [31]. The prepared adsorbent BC1 can make the food production and processing industry more efficient through niche applications. The adsorbent BC1 was designed with reusability in mind to facilitate its functioning within a circular economy. As will be shown in the remainder of this paper, BC1 has a mineral structure, which opens a wide avenue for further desorption and regeneration testing with various solvents or thermal treatments. This will also be the topic of our future publication. In this paper, a novel method of obtaining a nanostructured adsorbent material from waste blue crab shells is presented. The novel adsorbent, named BC1, is intended for adsorption of the pesticide acetamiprid.

## 2. Materials and Methods

### 2.1. Obtaining the Adsorbent Powder

Shells of the Atlantic blue crab (*Callinectes sapidus*) were gathered during regular environmental monitoring activities in the eastern Adriatic Sea (Croatia). The soft tissue was mechanically removed from the shells, carotenoids extracted, and micropowder produced according to the method recently described in detail by Nekvapil et al. [15].

Thermogravimetric analysis (TGA) of powdered carotenoid-depleted shells was conducted to establish the subsequent thermal treatment requirements. This was conducted using a TA Instruments SDT Q600 thermal analyzer (TA Instruments, New Castle, DE, USA) in the temperature range from 100 to 1100 °C. Separate runs were performed in air and in argon flow to find out how the presence or absence of oxygen in the atmosphere would affect the organic matter weight fraction and the onset of Mg-calcite decarboxylation.

The method of production of the adsorbent BC1 consisted of two main phases. In the first phase, the carotenoid-depleted shell powder was heated in argon flow to 500 °C for 2 h, followed by washing with NaOH aqueous solution (24% *w*/*w*). The aim of this phase was to remove any organic material that may block the nanopores or their walls. The second phase consisted of heating the powder in argon flow to 700 °C for 1.5 h, followed by HCl washing (1 M). The aim of this phase was to etch the walls of the nanopores and thereby increase the specific surface area of the calcite biomineral. The obtained BC1 adsorbent was kept in a drying oven at 105 °C for 2 h before use to ensure no atmospheric water molecules were adsorbed onto its surface.

### 2.2. Characterization of the Adsorbent Powder

Throughout this paper, reference is made to the two materials which yielded relevant analytical results. These are the intermediate blue crab shell products immediately after the 500 °C treatment and before the NaOH wash, termed BC-500, and the finalized adsorbent after the 700 °C treatment and HCl wash, termed BC1.

Specific surface area, pore volume, and their size distribution were measured using the procedure from Nekvapil et al. [15] via nitrogen sorption at 77 K using a Sorptomatic 1990 device (Thermo Electron, Waltham, MA, USA). Pre-treatment of the adsorbent precursor and the adsorbent at the final stage comprised degassing at 150 °C under vacuum for 4 h. The specific surface area was estimated according to the standard BET procedure (0.01–0.2 p/p_0_), while the total pore volume was estimated using the Dollimore–Heal method for the desorption branch.

Fourier transform infrared (FTIR) measurements were performed on a Perkin-Elmer model Spectrum BX II (Perkin-Elmer, Waltham, MA, USA). Spectra were recorded in the 400 to 4000 cm^−1^ range, with 1 cm^−1^ spectral resolution and 8 accumulation scans.

X-ray diffraction (XRD) analysis was conducted using a Bruker D8 Advance diffractometer (Bruker, MA, USA) equipped with a Cu source (λ_K_ = 0.15418 nm). Data were collected in the 10 to 60° 2θ range with steps of 0.03° 2θ.

### 2.3. Adsorption Experiments

During the adsorption process, the following parameters were optimized: pH (2–10), temperature (20–45 °C), adsorbent dose (0.2–1.2 g L^−1^), contact time (5–60 min), and initial pesticide concentration (10–60 g L^−1^). The adsorption was carried out in static mode and consisted of bringing into contact a solution of acetamiprid at a certain pH with the adsorbent in a Berzelius glass. The mixture was stirred at 300 rpm for a certain period of time and a certain temperature. At the end, the two phases were separated by centrifugation, and the acetamiprid was determined by high-performance liquid chromatography.

The degree of acetamiprid removal (η, %) on BC1 and the adsorption capacity (q_t_) of the adsorbent were determined from the following equations:(1)$$\eta \;\left(\%\right)=\frac{\left(C_{0}-C_{t}\right)}{C_{0}}100$$
(2)$$q_{t}=\left(C_{0}-C_{t}\right)\frac{V}{m}$$
where C_0_ and C_t_ (mg L^−1^) represent the acetamiprid concentration in the solution at the initial time and at time t (min), V (mL) is the volume of the acetamiprid solution, and m (g) is the amount of BC1.

The adsorption isotherm experiments were performed by stirring at 400 rpm 5 mg of BC1 with 5 mL of acetamiprid synthetic aqueous solutions having different initial concentrations (10–60 mg L^−1^) at 25 °C for 15 min.

### 2.4. Chromatographic Analysis of Acetamiprid

Analysis of acetamiprid from the aqueous solutions was performed using a LC2010 Shimadzu high-performance liquid chromatograph (Shimadzu, Kyoto, Japan) equipped with a diode array (DAD) detector. Acetamiprid was separated on an EC 250/4.6 NUCLEOSIL 100-5 C-18 column (Macherey-Nagel, Düren, Germany) and thermostated at 40 °C. Acetamiprid was isocratically eluted with acetonitrile and ultra-pure water with 0.1% formic acid (50:50, *v*/*v*) at a flow rate of 0.8 mL/min. A total of 5 μL of the water samples was passed through nylon syringe filters (0.45 μm × 13 mm) and injected, with three chromatograms being recorded for each sample.

## 3. Results

### 3.1. Obtaining the Adsorbent BC1 and Its Physical and Chemical Properties

TGA curves were recorded in both the air and argon atmospheres, in the range of 20–1100 °C (Figure 1). The curves show the same two major mass loss events: the first between about 245 and 385 °C, representing the decomposition of organic material, and the second between 625 and 750 °C, representing reversible decarboxylation of CaCO_3_ into CaO and CO_2_. The latter reaction starts slowly but progresses more rapidly as the temperature increases [32]. These events indeed exhibit shifts in the onset and ending temperatures between the two types of atmospheres. The total mass loss of the shell micropowder until 1100 °C was similar in both atmospheres, resulting in around 46% of the initial mass remaining. This indicates that in both cases, all of the organic matter had been volatilized, and calcite had been decarboxylated until reaching the end temperature. Indeed, chitin was shown to degrade in both the oxygen-containing and oxygen-free atmospheres but through different reaction mechanisms and temperature ranges. Carbon dioxide, carbon monoxide, and ammonia are the primary volatile products of chitin degradation in air, while acetamide and acetic acid are the products forming in an inert atmosphere [33,34]. Decarboxylation of CaCO_3_ would still occur in an argon atmosphere due to the presence of oxygen in the lattice, although at a slower pace. Here, we aim to exploit this reaction temperature offset to produce calcite even after treatment at 700 °C.

BET analysis of similar shell types published in our earlier paper [15] shows that native shell powder exhibits the specific surface area of 8.2 m^2^ g^−1^ and pore volume of 0.049 cm^3^ g^−1^, while the values were 32.9 m^2^ g^−1^ and 0.135 cm^3^ g^−1^, respectively, after the carotenoids extraction procedure. Here, a slight reduction in porosity was noted after heating at 500 °C in argon flow but before the NaOH washing to 27.59 m^2^ g^−1^ and 0.13 cm^3^ g^−1^. Ultimately, the finalized adsorbent powder BC1 exhibited the specific surface area of 250.38 m^2^ g^−1^ and pore volume of 0.4 cm^3^ g^−1^, with pore diameter distribution without a clear mean but overall, below 70 nm (Figure 2). The samples exhibited the type IV isotherms, as is usual for crab shell powder [15].

FTIR absorption spectra were recorded for the intermediate BC-500 and for the finalized adsorbent BC1 (Figure 3). The spectra of both samples show typical absorption bands of calcium carbonate at 712, 871, 1450, and 1795 cm^−1^. These correspond to (CO_3_^2−^) out-of-plane bending (also called the *v*_4_), *v*_2 asymm_ (CO_3_^2−^) and *v*_3 asymm_ (CO_3_^2−^), encompassing two modes (at 1421 cm^−1^, characteristic to crystalline calcite, and at 1473 cm^−1^, characteristic to amorphous CaCO_3_). The weak band at 1795 cm^−1^ corresponds to the harmonic of *v*_1_ (not visible) and *v*_4_ bands according to previously published assignments for extensively studied geological calcium carbonate [35,36]. Precise deconvolution of the *v*_3_ band is arguably difficult due to wide and overlapping characters of the component modes; however, major differences can be singled out in this region of BC-500 compared to BC1. BC-500 exhibits the crystalline mode clearly more strongly than the amorphous mode, similar to the situation encountered in native shells [15], while BC1 exhibits the amorphous 1473 cm^−1^ more strongly than the crystalline component. The other difference is in the wide band spanning the 2950 to 3700 cm^−1^ regions, assigned to O-H stretching (organic CH_2,3_ modes are not expected), arising from the structural water of ACC. This shows that the relative amount of amorphous CaCO_3_ phase increases as a result of basic, thermal (in argon flow), and acidic treatment.

To validate the bulk calcite and amorphous CaCO_3_ structure of BC1, i.e., the lack of CaO and Ca(OH)_2_ formation, supplementary tests were conducted. Here, the blue crab shell was heated at 800 °C for 1 h in the air atmosphere. The FTIR spectra of BC1 and the supplementary test are comparatively shown in Appendix A. The formation of oxide and hydroxide phases in the latter sample are visible by the red shift of the CO_3_^2−^ band and the appearance of the sharper O-H stretching mode of lime [37] at 3640 cm^−1^. The latter is otherwise absent in BC1 (Figure 3).

A previous study [15] reported the composition of the blue crab shell of similar origin to comprise a bulk magnesium-substituted calcite biomineral phase. Both stages currently being considered (i.e., BC-500 and finalized adsorbent BC1) exhibited characteristic reflection peaks of biogenic calcite. After treatment at 700 °C and then with HCl, the blue crab shell adsorbent showed the preserved trigonal structure of calcite (CaCO_3_). The amorphous CaCO_3_ signal was observed in FTIR spectra; however, because amorphous CaCO_3_, like other amorphous minerals, does not yield well-defined reflection peaks, it cannot be unambiguously pinpointed in XRD patterns (Figure 4). In the ambient air atmosphere, at such high temperatures, calcite would decarboxylate into CaO. However, the latter phase is very unstable and has a tendency to revert back to CaCO_3_ or Ca(OH)_2_ after lowering the temperature and being in contact with air humidity. Supplementary high-temperature tests show the distinct reflection peaks of lime (CaO) and portlandite [Ca(OH)_2_] that would form upon the treatment of crab shell powder in air at 800 °C after 1 h of treatment (Appendix A).

### 3.2. Investigation of Acetamiprid Adsorption on BC1

The amount of acetamiprid retained on the adsorbent was calculated using the calibration curve: y = 3 × 10^7^x − 6645.3 (R^2^ = 0.9999) drawn for the concentration domain 0.003−0.8 mg mL^−1^. The detection limit (LOD) and quantification limit (LOQ) determined using SMAC 2.0 software were 0.0018 mg mL^−1^ and 0.0033 mg mL^−1^, respectively.

The value of pH at the point of zero charge (pH_pzc_) was determined by the so-called pH drift method [38,39].

The pH_pzc_ value was studied using 0.01 M NaCl solution and determined by the intersect position of the pH 0 axis with the resulting measurement curve, as shown in Figure 5a. Also, the influence of the studied parameters (pH, temperature, adsorbent dose, time, and pesticide concentration) in the adsorption process on the degree of removal of acetamiprid on BC1 is shown in Figure 5b–f.

Following the study on the influence of pH (Figure 5b) on acetamiprid adsorption on BC1, it was found that the degree of pesticide removal varies with pH, the maximum value being recorded at pH 8. The optimal pH value established for acetamiprid removal is below the pH_pzc_ value and therefore the BC1 surface will have a positive charge and the acetamiprid anion can be adsorbed.

Infrared adsorption studies have confirmed that carbonic acid has the main influence upon the surface chemistry of calcium carbonate in normal atmospheric conditions. This is also true for aqueous solutions, albeit the carbonic acid here rapidly dissolves into CO_2_ and H_2_O [39].

The solubility of calcium carbonate increases as the pH becomes increasingly acidic [40,41]. When the pH = 8, at which the maximum acetamiprid removal rate is observed, this is below the point of zero charge (pH_pzc_ = 9.14), which makes its surface positively charged and hence favorable for acetamiprid adsorption from an electrochemical point of view. The solubility of calcium carbonate (and BC1 itself) increases as the pH becomes increasingly acidic. Thus, a pH of 8 is still above the pH value where notable dissolution of CaCO_3_ occurs, which accounts for the lower and erratic removal rate. In fact, below this pH, the calcite reacts with HCl from the solution, a part of this being transformed into CaCl_2_, a soluble salt. Due to this fact, the adsorption can vary and the equilibrium is unstable at a pH less than 8. On the other hand, at an acidic pH, the acetamiprid molecule will be protonated and the interactions between CaCO_3_ and acetamiprid are modified. The variation of removal degree is due to cycles of adsorption–desorption.

The results obtained from the study of the effect of temperature on the adsorption process (Figure 5c) showed that the degree of acetamiprid removal increases greatly from 20 °C to 25 °C, then it varies insignificantly with the increase in temperature up to 35 °C, after which it decreases greatly at temperatures higher than 40 °C. The solubility of both CO_2_ and CaCO_3_ in water decreases with increasing solution temperature [40,41]. This indirectly influences the carbonic acid formation and dissolution. The range between 25 and 35 °C may be an anomalous interval where the carbonate chemistry is strongly favorable for acetamiprid adsorption. However, the current study cannot provide a definite answer for the underlying process.

The degree of removal increases with the increase in the adsorbent dose (Figure 5d), reaching 90.68% for 1.2 g L^−1^ of BC1. Since the increase in the degree of acetamiprid removal from 1 g L^−1^ to 1.2 g L^−1^ is not significant and for economic reasons, the dose of BC1 of 1 g L^−1^ was used for the subsequent studies.

As can be seen in Figure 5e, a rapid increase in the removal rate of acetamiprid can be observed in the first 15 min, after which a decrease is recorded.

The observations regarding removal rates at different contact times could be explained through the fact that due to the complex motions of the molecules within the solution, both adsorption and desorption processes occur. Increasing removal rates are observed at contact times until the first 15 min due to the great availability of surface area for adsorption. Removal rates seem to decrease at contact times from 15 to 30 min due to the higher desorption rate resulting from reaching and possibly breaching of the BC1 capacity. At longer contact times, a balance between the two processes seems to be reached, whereby a similar amount of acetamiprid is adsorbed at any given moment.

The effects of the quantity of BC1 and the initial pollutant dose can also be explained through the availability of the adsorption surface area on BC1.

Taking into account the obtained results, 20 mg L^−1^ was the optimal initial concentration of acetamiprid (Figure 5f), at which the highest degree of removal was obtained. Initially, with the increasing pesticide concentration, the degree of removal increases and then decreases, which is probably due to the fact that the active centers on the adsorbent are occupied.

The optimum conditions for the removal of acetamiprid on BC1 were found to be as follows: 20 mg L^−1^ of initial pesticide concentration, pH 8, 1 g L^−1^ of adsorbent dose, 25 °C, and 15 min contact time. In these conditions, the adsorption capacity of the adsorbent was 17.8 mg g^−1^.

### 3.3. Adsorption Isotherm

The experimental data from the initial concentration of acetamiprid’s influence on removal degree were evaluated using the linearized equations of Langmuir and Freundlich isotherm models [42].

The Langmuir isotherm considers that the adsorption is monolayer, because the adsorption centers are the same, while the Freundlich isotherm is an empirical equation and it is applicable for multilayer sorption [43,44]. The parameters determined from the two models are shown in Table 1.

By comparing the two models, it can be seen that the Langmuir isotherm best fits the adsorption of acetamiprid on BC1 because in this one, the coefficient of determination R^2^ has the value closest to unity. Therefore, the adsorption of acetamiprid on BC1 is monolayer.

The two isotherms applied to the adsorption of acetamiprid on BC1 were compared with the experimental data obtained at 25 °C (Figure 6) in order to determine the isotherm which is best suited.

## 4. Discussion

It could be argued that the removal of chitin affected the absorption properties of the nanoporous shell powder. Indeed, chitin and its derivative, chitosan, were previously shown to have an adsorption capacity for a range of substances, as reviewed recently by da Silva Alves et al. [45] and demonstrated by Fabbricino et al. [46] and Rissouli et al. [47]. Calcite, on the other hand, received more attention in the context of metal and ion adsorption [30,43,44]. However, it is argued here that exploiting chitin fibers may not be the best use of nanoporous crab shell biominerals in all cases. Again, the importance of detailed knowledge of the shell structure of particular crustacean species is re-iterated in order to choose the best pathway for valorization. Our TGA results indicate that the sample’s loss of mass in the 270–370 °C range where the chitin was degraded amounted to about 14%, indicating that this polymer is a minor component of the blue crab shell. On the other hand, through presented treatments, the porosity of BC1 nanoporous powder increased by about 32 times relative to the native material. This translates to a substantially larger free mineral surface area for contact with pollutants.

Methods for obtaining adsorbents for acetamiprid were also explored for a variety of other waste-derived biomass types. The most relevant ones and their main properties, which were consistently reported, are summarized in Table 2. Though some properties are intrinsic to the adsorbent, such as specific surface area and pH_pzc_, other parameters, such as pollutant removal efficiency assuming various initial adsorbent or pollutant concentrations, are dependent on the particular experimental setup. The capacities of certain bio-derived adsorbents for retaining common dyes were also summarized by Inthapanya et al. [48]. Prepared adsorbents, such as MOFs, were also designed for the removal of insecticides. Although these are synthesized and currently cannot be considered as significantly improving waste management, one example is included in Table 2. For instance, Liu et al. [49] produced a magnetic graphene oxide-β-cyclodextrin metal–organic framework and showed its absorption capacities for several neonicotinoid pesticides, including acetamiprid. This adsorbent featured a specific BET surface area of 250.33 m^2^ g^−1^, similar to our BC1 presented here, and exhibited the adsorption capacity for acetamiprid of 2.96 mg g^−1^ at an insecticide concentration of 100 mg L^−1^ in a static adsorption system.

The pH value is one of the main factors in adsorption trials as it influences the ionic states of the adsorbent surface and the adsorbate. However, it is important to observe that the same adsorption target may have different affinities relative to the adsorbent. For instance, Mohammad et al. [50] and Sahroui et al. [51] presented an activated carbon adsorbent which takes up acetamiprid most efficiently at an acidic pH. Being of biomineral origin, our adsorbent BC1 exhibits a point of zero charge around pH 9, with the most efficient tested adsorption of acetamiprid at pH 8, which allows the use of BC1 in neutral to slightly basic conditions. Crab shell biochar exhibits a neutral pH_pzc_ [24]; however, the cited study presents well the dependence of pH of the highest efficiency also on the ionic affinity of the target dye. Mohammad et al. [50] reported the highest surface area and acetamiprid adsorption capacity for an activated carbon based adsorbent. Hence, the specific surface area is one of the main determinators of the adsorption capacity. Overall, different waste-derived adsorbents with varying properties allow certain flexibility for applicative purposes.

## 5. Conclusions

The method of obtaining of a novel nanoporous biomaterial from waste crab shells (BC1) is presented, and the obtained adsorbent was characterized by different techniques. Also, the considerations regarding choosing the type of shell for the starting material are given, allowing true knowledge-based action in the spirit of the blue bioeconomy.

The efficacy of the novel adsorbent was tested for acetamiprid removal from aqueous solutions, taking into account adsorption parameters such as pH, temperature, contact time, initial concentration, and adsorbent dose. The optimum conditions at which the adsorption of acetamiprid on BC1 takes place are an initial concentration of acetamiprid of 20 mg L^−1^, a pH 8, 1 g L^−1^ of BC1, a temperature of 25 °C, and a 15 min contact time. Under these conditions, the adsorption capacity of BC1 was 17.8 mg g^−1^.

Langmuir and Freundlich isotherm models were used for predicting the acetamiprid adsorption on the prepared adsorbent. It was concluded that the equilibrium data fitted well with the Langmuir model, according to which the maximum adsorption capacity was 26.455 mg g^−1^.

According to the results obtained in this study, it can be concluded that the obtained novel adsorbent BC1 could be used to remove acetamiprid from aqueous solutions. Moreover, as the BC1 nanoporous adsorbent is in the form of a micropowder, this allows further development of filter designs and niche applications to support the development of sustainable waste management and water purification.

## Figures and Tables

**Figure 1 nanomaterials-13-03042-f001:**
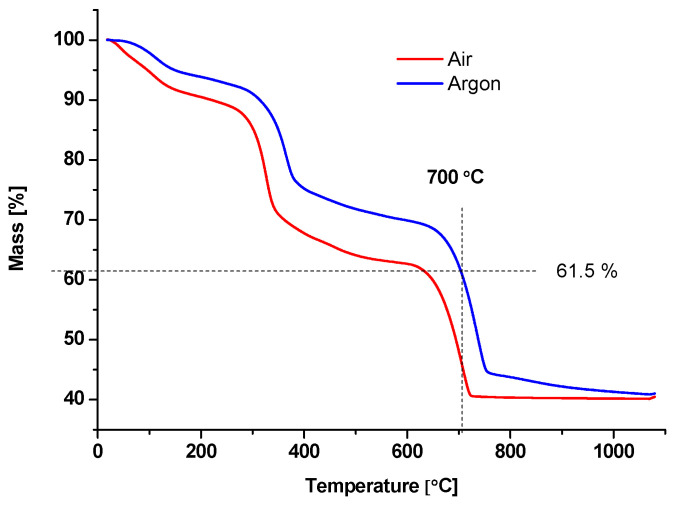
Thermogravimetric analysis (TGA) of the Atlantic blue crab (*Callinectes sapidus*) shell powder. Analyses were conducted for the same powder stock, with separate runs in normal air atmosphere and in argon flow. Key comparative points are labeled on the graph: the 700 °C on the *X*-axis, and the final sample mass of 61.5% on the *Y*-axis.

**Figure 2 nanomaterials-13-03042-f002:**
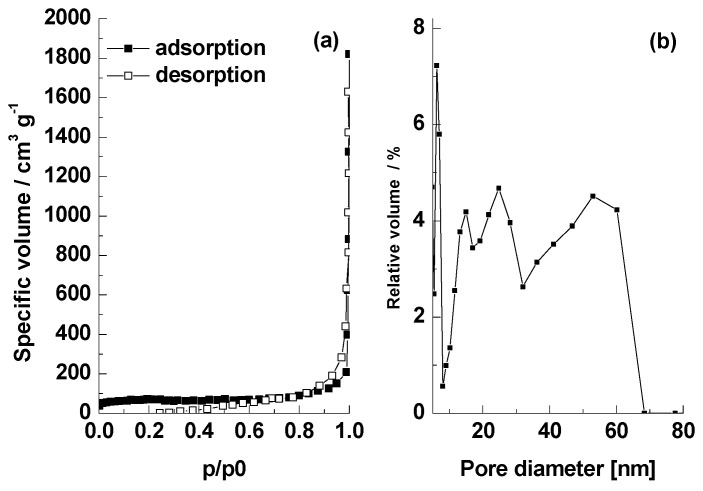
Nitrogen physisorption (BET method) results for BC1 adsorbent: (**a**) adsorption and desorption isotherms from which BET surface area and pore volume were calculated; (**b**) pore diameter distribution.

**Figure 3 nanomaterials-13-03042-f003:**
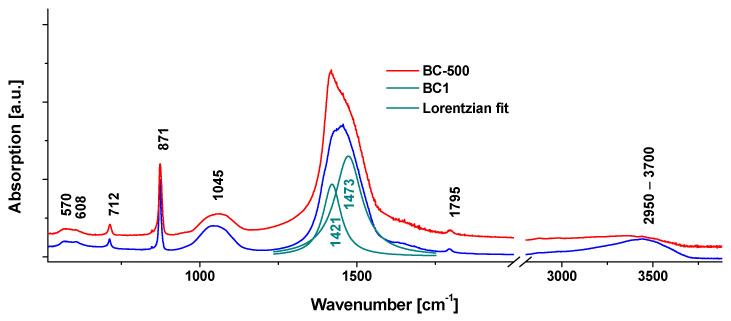
FTIR spectra of the adsorbent BC1 and an intermediary step analyzed after 500 °C (BC-500) obtained from the Atlantic blue crab (*Callinectes sapidus*) shell powder. Spectra have been normalized to the sharp calcite band at 871 cm^−1^ and y-offset was applied for clarity. Lorentzian multi-peaks fit (green bands) refers to BC1 spectrum.

**Figure 4 nanomaterials-13-03042-f004:**
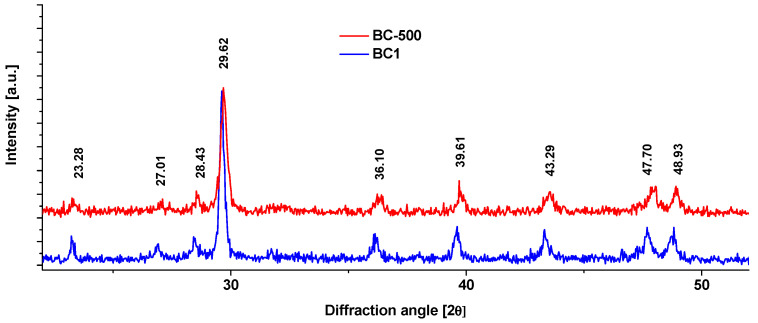
XRD patterns of the adsorbent BC1 and an intermediary step analyzed after 500 °C (BC-500) obtained from the Atlantic blue crab (*Callinectes sapidus*) shell powder. Y-offset was applied for clarity.

**Figure 5 nanomaterials-13-03042-f005:**
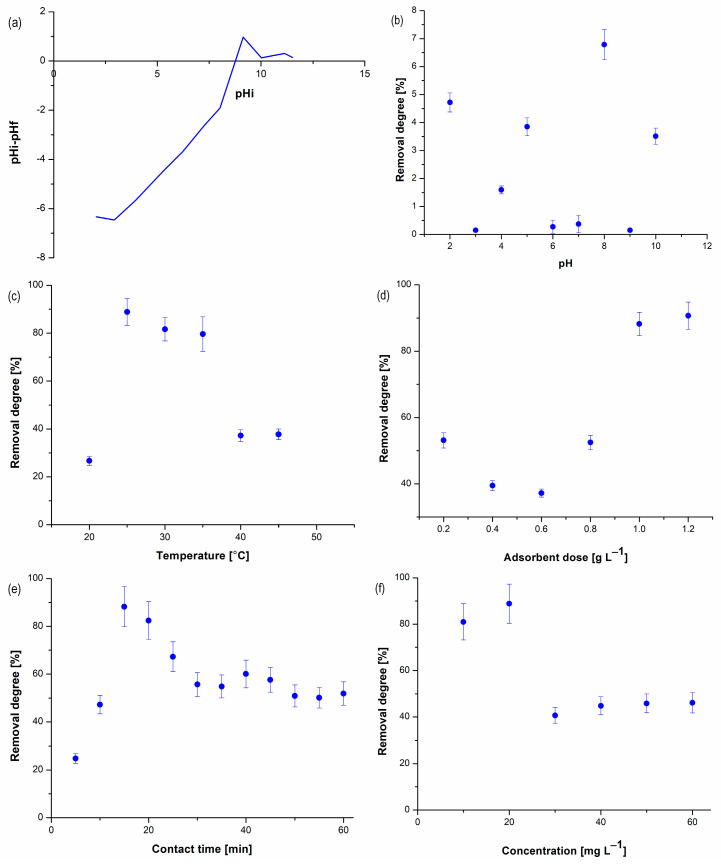
Point zero charge of BC1 (**a**) and the effect of the pH (20 mg L^−1^, 25 °C, 15 min, 1 g L^−1^ BC1) (**b**), the temperature (20 mg L^−1^, pH 8, 15 min, 1 g L^−1^ BC1) (**c**), the adsorbent dose (20 mg L^−1^, pH 8, 25 °C, 15 min) (**d**), the contact time (20 mg L^−1^, pH 8, 25 °C, 1 g L^−1^ BC1) (**e**), and the initial concentration of acetamiprid (pH 8, 25 °C, 15 min, 1 g L^−1^ BC1) (**f**) on adsorption process. Each data point is the mean ± the standard error of the mean of three independent experiments.

**Figure 6 nanomaterials-13-03042-f006:**
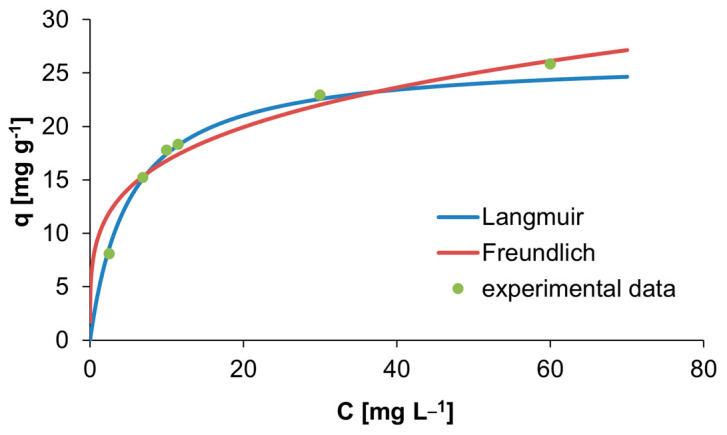
Isotherms of acetamiprid adsorption on BC1 at 25 °C.

**Table 1 nanomaterials-13-03042-t001:** Isotherm parameters for the adsorption of acetamiprid on BC1.

Isotherm Model	Constants	Values
Langmuir	q_m_ [mg g^−1^]	26.4550
	K_L_ [L g^−1^]	0.1927
	R^2^	0.9123
Freundlich	K_F_ [L mg^−1^]	9.5258
	1/n	0.2463
	R^2^	0.5862

**Table 2 nanomaterials-13-03042-t002:** Presentation of some of the previously explored adsorbents, either obtained from calcium carbonate biomaterials or used for acetamiprid adsorption.

Starting Material	Composition	Pollutant	Adsorption Capacity (mg g^−1^)	References
Oyster shells	CaCO_3_	Acid green	33.3	[45]
Uca crab	Native shell (CaCO_3_ and chitin)	Reactive blue 222	4.19	[14]
Unspecified crab shell stock	biochar	Phosphorus	n.r.	[23]
Unspecified crab shell stock	Biochar	Malachite green; Congo red	12,501.98; 20,317.47	[24]
Unspecified crab shell stock	Biochar	Malachite green	6142.5	[25]
Unspecified crab shell stock	Native shell (CaCO_3_ and chitin)	Organochlorine pesticides	0.001 to 0.0015 for 4,4’-DDE	[27]
Graphene oxide-β-cyclodextrin metal–organic framework (MOF)	Magnetic MOF	Acetamiprid	2.96	[49]
Tangerine peel	Activated carbon	Acetamiprid	35.7	[50]
Pomegranate waste	Activated carbon	Acetamiprid	27.62	[51]
Eucalyptus woodchip	Biochar	Acetamiprid	4.87	[52]
BC1	CaCO_3_	Acetamiprid	26.45	This study

## Data Availability

Data are contained within the article and Appendix A.

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
