# Peer review of "A Novel Nanoporous Adsorbent for Pesticides Obtained from Biogenic Calcium Carbonate Derived from Waste Crab Shells"

_nanomaterials, 2023, doi:10.3390/nano13233042_

Round 1

Reviewer 1 Report

Comments and Suggestions for Authors

The study suitably describes the preparation of a novel nanoporous material prepared from the thermal treatment and chemical wash of crab shells. After a full characterization, the novel material is applied to the retention of a pesticide (acetamiprid). One of the main concerns of this study is that is only applied to the retention of one single compound, and it would be much more interesting to prove the suitable retention (or not) of different compounds. Another concern is about the validation. I am not familiar with the SMAC software; however, there is no point to report a LOQ concentration higher than the lowest point of the calibration curve since the calibration curve is intended to quantify. Please, revise these values. In addition, in the introduction, the paragraph in lines 107-113 is not linked and it might be moved after line 96.

Author Response

The authors thank the reviewer for relevant comments which improved the quality of our paper.

Comment: The study suitably describes the preparation of a novel nanoporous material prepared from the thermal treatment and chemical wash of crab shells. After a full characterization, the novel material is applied to the retention of a pesticide (acetamiprid). One of the main concerns of this study is that is only applied to the retention of one single compound, and it would be much more interesting to prove the suitable retention (or not) of different compounds.

Answer: We acknowledge the concern. The scope of the current paper was to present the BC1 adsorbent: the method for obtaining it and showing the proof of concept that a nanoporous calcium carbonate adsorbent obtained from crab shell food waste does indeed exhibit adsorptive capabilities. As indicated in the answer to a similar observation from another reviewer, we are committed to further developing the depollution method centered around the BC1 adsorbent, and a dedicated paper with a different scope will present robust adsorption and regeneration experiments using a range of additional pollutants (e.g. glyphosate, drugs etc). Hence, we preferred to structure the current paper with the scope of the special issue, presenting the adsorbent and its effectiveness.

Comment: Another concern is about the validation. I am not familiar with the SMAC software; however, there is no point to report a LOQ concentration higher than the lowest point of the calibration curve since the calibration curve is intended to quantify. Please, revise these values.

Answer: We checked and corrected.

Comment:  In addition, in the introduction, the paragraph in lines 107-113 is not linked and it might be moved after line 96.

Answer: Thank you for this good observation. We have moved the paragraph as suggested.

Reviewer 2 Report

Comments and Suggestions for Authors

The manuscript deals to produce nanoporous adsorbent from a food wastes, namely crab shells. The adsorbents were then characterized and adsorption experiments were performed. The subject is interesting and up-to date, it is related to the profile of the journal.  The structure of the manuscript is generally clear, several up-to date methods were used; however, the method of determination of concluded optimal condition isn’t adequate; it requires more sophisticated statistical analysis. The English of the manuscript is good, few spelling mistakes can be found, please check it carefully.

 Comments and questions:

 The Title is informative.

  1. The Abstract reflects the approach of the study, it summarizes the findings of the work.
  2. The section Introduction presents the important points of the topic, it contains references related to the earlier results, reveals the importance and originality of the work. However, it also should be emphasized, that how these method can be fitted into the concept of circular economy (as it is mentioned in line 115). What will be the fate of the adsorbent after adsorption? Can it be regenerated?
  3. Materials and methods: The experimental design is appropriate and adequately described.
  4. Results and discussion: In this section, authors describe the results shown in the corresponding figures, and tables. The figures generally are nice, but in some cases the figures must be improved.
    • Fig 5. Generally, these measured points are discrete values, using continuous lines are misleading. Moreover, standard deviations also should be given; e.g. in case of fig 5.b. the differences may not be significant. Please, label the circumstances in all cases in the figure title (e.g. adsorbent dose, temperature, concentration, contact time should be given in Fig. 5.b., etc. All figures must be self-explaining.
    • Adsorption mechanism, as was this monolayer or multilayer adsorption in the function of concentration, determination of adsorption isotherm etc. also should be investigated.
    • It is not clear, how did you determined the optimal circumstances; for this a more sophisticated statistical analysis should be performed.
  5. Discussion section doesn’t discuss the results, just compare the results to others’ results. It is very useful, but it would be more visible if summarized in a table.
  6. The Conclusions section summarizes shortly the result of the work.
  7. Conclusion and recommendation: This manuscript is recommended for publication after major revision.

Author Response

The authors thank to reviewer for relevant comments which improved the quality of our paper.

The manuscript deals to produce nanoporous adsorbent from a food wastes, namely crab shells. The adsorbents were then characterized and adsorption experiments were performed. The subject is interesting and up-to date, it is related to the profile of the journal.  The structure of the manuscript is generally clear, several up-to date methods were used; however, the method of determination of concluded optimal condition isn’t adequate; it requires more sophisticated statistical analysis. The English of the manuscript is good, few spelling mistakes can be found, please check it carefully.

Comments and questions:

Comment: The Title is informative.

Comment: The Abstract reflects the approach of the study, it summarizes the findings of the work.

Comment: The section Introduction presents the important points of the topic, it contains references related to the earlier results, reveals the importance and originality of the work. However, it also should be emphasized, that how these method can be fitted into the concept of circular economy (as it is mentioned in line 115). What will be the fate of the adsorbent after adsorption? Can it be regenerated?

Answer: We have added a text section at the end of the Introduction, as follows: “The adsorbent BC1 was designed with reusability in mind to address one of the three R`s of waste management within the circular economy (reduce, reuse, recycle). As will be shown in the remainder of this paper, the BC1 has a mineral structure, which opens a wide avenue for further desorption and regeneration testing with various solvents or thermal treatments, which will be the topic of our future publication.” We are indeed committed to developing a novel decontamination method centered around the adsorbent BC1, and we are working on regeneration tests with multiple solvents (including EtOH, EDTA, NaOH, HCl and H2SO4), and these tests include a wider range of pollutants. However, we envisaged these tests for a future paper with a specific scope targeting a robust regeneration study.

Materials and methods:

Comment: The experimental design is appropriate and adequately described.

Results and discussion:

Comment: In this section, authors describe the results shown in the corresponding figures, and tables. The figures generally are nice, but in some cases the figures must be improved.

Fig 5. Generally, these measured points are discrete values, using continuous lines are misleading. Moreover, standard deviations also should be given; e.g. in case of fig 5.b. the differences may not be significant. Please, label the circumstances in all cases in the figure title (e.g. adsorbent dose, temperature, concentration, contact time should be given in Fig. 5.b., etc. All figures must be self-explaining.

Answer: Done.

Comment: Adsorption mechanism, as was this monolayer or multilayer adsorption in the function of concentration, determination of adsorption isotherm etc. also should be investigated.

Answer: The adsorption isotherm was added at point 3.3.

It is not clear, how did you determined the optimal circumstances; for this a more sophisticated statistical analysis should be performed.

Answer: The optimal adsorption conditions were determined one by one for each parameter. For e.g., pH was varied between 2 and 10 keeping the other parameters constant. Since the highest degree of adsorption was obtained at pH 8, in the following experiments the pH of the acetamiprid solution was brought to pH 8.

When optimizing the temperature, the temperature was varied between 20 and 45 °C keeping the other parameters constant, with pH at 8, which was determined previously. The best value was determined at 25 °C, the value at which the following experiments will be performed.

When determining the adsorbent dose, its value varied between 0.2 and 1.2 g L-1 keeping other parameters constant, with pH at 8 and temperature at 25 °C, values determined previously.

On the same principle, the optimal contact time and initial concentration of acetamiprid were determined.

Comment: Discussion section doesn’t discuss the results, just compare the results to others’ results. It is very useful, but it would be more visible if summarized in a table.

Answer: We have created a table (Table 1) summarizing the relevant published literature, and modified the Discussion chapter referring to the new table. The modifications are not specifically highlighted because they are many and rather dense.

Comment: The Conclusions section summarizes shortly the result of the work.

Conclusion and recommendation: This manuscript is recommended for publication after major revision.

Reviewer 3 Report

Comments and Suggestions for Authors

Review

Journal: Nanomaterials

Manuscript ID: nanomaterials-2727345

Title: A novel nanoporous adsorbent for pesticides obtained from biogenic calcium carbonate, derived from waste crab shells

The authors report the preparation of an adsorbent using crab shell waste. They investigated the effects of pH, temperature, adsorbent dose, contact time, and pollutant concentration on the adsorption of a pesticide (acetamiprid). Unfortunately, my opinion of the paper is mostly negative. Although the main idea and its execution are certainly commendable and the introduction is well-constructed (apart from grammatical errors), the discussion of the results is severely limited. The most important figure in the manuscript is Figure 5, which shows the effect of different parameters on the adsorption. The results do not show trend-like changes in any of the cases, which is not a problem in itself, but the fact that virtually no explanation is provided is certainly problematic. Section 4, which is the discussion section, is a collection of literature examples loosely related to the work presented. It does not contain a discussion of the results shown in this work. The English of the paper should also be significantly improved by a proofreader. Based on this, although I am sorry, I cannot recommend the publication of this work in its present form in Nanomaterials. I suggest resubmitting the work after a major overhaul. My suggestions are as follows:

Scientific comments:

1)     The authors should explain why both air and Ar atmospheres were used for the TG measurements. What additional information (related to the intended usage of the adsorbents) did these measurements provide? Moreover, the fact that the weight loss was higher for the air atmosphere than for the Ar atmosphere should be correlated with the presence/absence of oxygen.

2)     Line 236: What can be the reason that BC-500 exhibits the crystalline mode stronger than the amorphous mode in comparison to BC1? Higher temperatures facilitate crystallization, and the preparation of BC1 involved another heat treatment at a higher temperature.

3)     Line 263: Only crystalline materials yield (well-defined) diffractions in XRD measurements. It is thus plausible that amorphous CaCO3 was detectable by IR measurements, but not detectable by XRD measurements. The sentence should be changed accordingly.

4)     The chemical formula Ca(OH)2 should be used instead of CaOH and Ca(OH).

5)     When applying offsets (Fig. 4), it is more conventional not to include the y-axis numbers, and instead, write “Intensity (a.u.)” as the y-axis title, standing for “arbitrary units”.

6)     The results of Figure 5 should be discussed in detail. For example, in Fig. 5b, what can be the reason that in the acidic domain (e.g., between pH 2 and pH 6) the adsorption first decreases, then increases, then decreases again?

7)     In Fig. 5c, what can be the reason that adsorption increases from ~20% to ~90% because of increasing the temperature merely by five degrees? This is one of the most astonishing results of this work, which must be explained.

8)     In Fig. 5d, what can be the reason that adsorption increases from ~50% to ~90% because of slightly increasing the adsorbent dose from ~0.9 g/L to ~1.1 g/L? Changing the adsorbent dose by a similar margin did not result in such a dramatic change before and after this threshold. Why?

9)     Error bars should be included in Fig. 5.

10) The adsorption capacity obtained in this work should be compared with literature data. A table should be prepared based on the discussion section (Section 4), showing how the adsorption capacity changes as a function of different starting materials and parameters. What do the authors think, how reliably can the adsorption capacities from different starting materials be compared to each other?

11) The conclusion section should be revised. Currently, it does not contain any specific results. In terms of their intended role, currently, the abstract section would serve better as the conclusion section and vice versa.

Comments on the Quality of English Language

English- and writing-related comments:

1)     The use of the definite article “the” is erroneous in many cases. For example, they should not precede plural, general nouns.

2)     There are many punctuation mistakes, especially when modal verbs are used. The manuscript contains many prepositional mistakes too.

3)     Many sentences are too long. For example, Lines 79–93 contain three sentences that are over 50 words long. An average sentence should not be longer than 20–25 words, to retain intelligibility.

4)     Very often the main verb of the sentence is also the last word of the sentence, which hinders intelligibility.

5)     Because the sentences are long, parallel structure is used in many cases. However, the rule for constructing correct parallel structures is not followed.

6)     The rules of economic writing should be followed. Periphrastic constructions should be avoided. For example, the sentence in Lines 52–54 contains so many mistakes that most of the problems I listed above can be found there. The sentences in Lines 257–260 are also good examples.

7)     There are sentences that are incomplete (Lines 42–44), contain more than one predicate (Lines 364–368), or are incomprehensible (Lines 130–131).

8)     Expressions such as “high-quality analytical results” (Line 146) should be avoided. Readers expect the results to be reliable either way.

9)     The authors should highlight better when they refer to their previous work. For example, in Line 212, instead of “BET analysis of similar material presented in the paper by Nekvapil et al. [15] shows that”, the authors should write “BET analysis of a similar material presented in our previous paper [15] shows that” to avoid possible confusions.

10) Line 287: The correct word to use in this context is “intersects”, not “cuts”.

Author Response

The authors thank the reviewer for relevant comments which improved the quality of our paper.

The authors report the preparation of an adsorbent using crab shell waste. They investigated the effects of pH, temperature, adsorbent dose, contact time, and pollutant concentration on the adsorption of a pesticide (acetamiprid). Unfortunately, my opinion of the paper is mostly negative. Although the main idea and its execution are certainly commendable and the introduction is well-constructed (apart from grammatical errors), the discussion of the results is severely limited. The most important figure in the manuscript is Figure 5, which shows the effect of different parameters on the adsorption. The results do not show trend-like changes in any of the cases, which is not a problem in itself, but the fact that virtually no explanation is provided is certainly problematic. Section 4, which is the discussion section, is a collection of literature examples loosely related to the work presented. It does not contain a discussion of the results shown in this work. The English of the paper should also be significantly improved by a proofreader. Based on this, although I am sorry, I cannot recommend the publication of this work in its present form in Nanomaterials. I suggest resubmitting the work after a major overhaul. My suggestions are as follows:

Scientific comments:

Comment: The authors should explain why both air and Ar atmospheres were used for the TG measurements. What additional information (related to the intended usage of the adsorbents) did these measurements provide? Moreover, the fact that the weight loss was higher for the air atmosphere than for the Ar atmosphere should be correlated with the presence/absence of oxygen.

Answer: The sections referring to TGA in Materials and methods and Results sections were improved for better clarity and explanations.

 Comment: Line 236: What can be the reason that BC-500 exhibits the crystalline mode stronger than the amorphous mode in comparison to BC1? Higher temperatures facilitate crystallization, and the preparation of BC1 involved another heat treatment at a higher temperature.

Answer: After the last higher thermal treatment, at 700 °C, the material was washed with HCl intended to etch the pore walls. The last treatment acted to dissolve parts of the mineral phase and organic residues, leaving behind an amount of amorphous CaCO3. A phrase was added in the Materials and Methods section where the samples are introduced, to clarify that the HCl wash was done before FTIR measurements of BC1.

Comment: Line 263: Only crystalline materials yield (well-defined) diffractions in XRD measurements. It is thus plausible that amorphous CaCO3 was detectable by IR measurements, but not detectable by XRD measurements. The sentence should be changed accordingly.

Answer: We thank the Reviewer for pointing out this issue, and consequently we were able to improve the clarity of the XRD section in the Results part.

Comment: The chemical formula Ca(OH)2 should be used instead of CaOH and Ca(OH).

Answer:  The corrections were applied.

Comment: When applying offsets (Fig. 4), it is more conventional not to include the y-axis numbers, and instead, write “Intensity (a.u.)” as the y-axis title, standing for “arbitrary units”.

Answer: We have applied the Reviewer`s suggestion in all figures that include y-offset in the main manuscript and also in supplementary material.

Comment: The results of Figure 5 should be discussed in detail. For example, in Fig. 5b, what can be the reason that in the acidic domain (e.g., between pH 2 and pH 6) the adsorption first decreases, then increases, then decreases again?

In Fig. 5c, what can be the reason that adsorption increases from ~20% to ~90% because of increasing the temperature merely by five degrees? This is one of the most astonishing results of this work, which must be explained.

In Fig. 5d, what can be the reason that adsorption increases from ~50% to ~90% because of slightly increasing the adsorbent dose from ~0.9 g/L to ~1.1 g/L? Changing the adsorbent dose by a similar margin did not result in such a dramatic change before and after this threshold. Why?

Answer: Some details were added in the Discussions section, but it is not possible to explain now the variation of the efficiency when we varied the parameters with small values. This is a preliminary study for proving the possibility to retain pesticides from contaminated water on calcite. Also, to the best of our knowledge, the literature on organic pollutants removal using CaCO3 adsorbents it is almost completely missing. The existing literature largely focused on removal of ionic pollutants (mostly P-containing compounds and heavy metals), which may have different adsorption mechanisms and thus are not comparable in the context of our study.. Also, by default, the optimization studies on the retention of organic pollutants on this material are missing. We will consider all these observations for the future elaborated studies that we will perform on this material.

Comment: Error bars should be included in Fig. 5.

Answer: Added.

Comment: The adsorption capacity obtained in this work should be compared with literature data. A table should be prepared based on the discussion section (Section 4), showing how the adsorption capacity changes as a function of different starting materials and parameters. What do the authors think, how reliably can the adsorption capacities from different starting materials be compared to each other?

Answer: We have created a table (Table 1) summarizing the relevant published literature, and modified the Discussion chapter to reference the new table. Overall, we agree with the Reviewer on the necessity of such a comparison, however, this is hampered by the lack of standardized format of presentation of the adsorption results. In many cases the content heavily depends on the background of the research team; where the study is led by physicists/material scientists, the focus is on the technical properties of the adsorbent, while in the case of a team led by adsorption scientists, the discussion is weighted towards complete exploration of the adsorption. Nevertheless, we have attempted to compile the relevant data from articles related to our topic. The modifications are not specifically highlighted because they are many and rather dense.

Comment: The conclusion section should be revised. Currently, it does not contain any specific results. In terms of their intended role, currently, the abstract section would serve better as the conclusion section and vice versa.

Answer: Conclusions section was revised.

Comments on the Quality of English Language

General answer to comments regarding English language:  We acknowledge that there were English language errors. We have screened again the manuscript with the help of a proficient English speaker. Longer sentences were shortened and the grammar was improved. We did the revision using TrackChanges, so the modifications are visible.

English- and writing-related comments:

The use of the definite article “the” is erroneous in many cases. For example, they should not precede plural, general nouns.

There are many punctuation mistakes, especially when modal verbs are used. The manuscript contains many prepositional mistakes too.

Many sentences are too long. For example, Lines 79–93 contain three sentences that are over 50 words long. An average sentence should not be longer than 20–25 words, to retain intelligibility.

Very often the main verb of the sentence is also the last word of the sentence, which hinders intelligibility.

Because the sentences are long, parallel structure is used in many cases. However, the rule for constructing correct parallel structures is not followed.

The rules of economic writing should be followed. Periphrastic constructions should be avoided. For example, the sentence in Lines 52–54 contains so many mistakes that most of the problems I listed above can be found there. The sentences in Lines 257–260 are also good examples.

There are sentences that are incomplete (Lines 42–44), contain more than one predicate (Lines 364–368), or are incomprehensible (Lines 130–131).

Answer: Revised.

Comment: Expressions such as “high-quality analytical results” (Line 146) should be avoided. Readers expect the results to be reliable either way.

Answer: This was the mistake of word choice. We have replaced “high-quality” with “relevant”. The intention was to reference comparatively more important results and those featuring better signal quality.

Comment: The authors should highlight better when they refer to their previous work. For example, in Line 212, instead of “BET analysis of similar material presented in the paper by Nekvapil et al. [15] shows that”, the authors should write “BET analysis of a similar material presented in our previous paper [15] shows that” to avoid possible confusions.

Answer: The reviewer is right, this sentence was unclear. We applied the correction, and we used the word “shell type” instead of “material”.

Comment: Line 287: The correct word to use in this context is “intersects”, not “cuts”.

Answer: The suggestion was implemented.

Round 2

Reviewer 2 Report

Comments and Suggestions for Authors

The manusript significantly improved according to the reviewer's requirements. However, the adsorption isotherms part should be elaborated in more detailed. Please add the method to experimental section; moreover, the fitted data also  should be presented in a figure. R2 also should be added to the table. The statement, (line 463) "while the Freundlich isotherm assumes that the adsorption is multilayer due to the unlimited number of available centers [40,41]." isn't valid; both model represents monolayer adsorption. References don't support your statement.

Now it is recommended for publication after miron revision. 

Author Response

Dear reviewer,

Thank you for your comments!

You can find our answers below and we hope that our paper is in accordance with your requirements, now.

Best regards,

Loredana Soran

The manusript significantly improved according to the reviewer's requirements. However, the adsorption isotherms part should be elaborated in more detailed. Please add the method to experimental section; moreover, the fitted data also should be presented in a figure.

Answer: Added.

 R2 also should be added to the table.

Answer: R2 it is already added in Table 2.

The statement, (line 463) "while the Freundlich isotherm assumes that the adsorption is multilayer due to the unlimited number of available centers [40,41]." isn't valid; both model represents monolayer adsorption. References don't support your statement.

Answer: The fragment "while the Freundlich isotherm assumes that the adsorption is multilayer due to the unlimited number of available centers [40,41]." was rewritten as “"while the Freundlich isotherm is an empirical equation ant it is applicable for multilayer sorption [43,44]." The number of references were changed.

The two references were changed with:

Gimbert, F.; Morin-Crini, N.; Renault, F.; Badot, P.M.; Crini, G. Adsorption isotherm models for dye removal by cationized starch-based material in a single component system: Error analysis. J. Hazard. Mater. 2008, 157, 34–46.

Mahmoud, H.R.; Ibrahim, S.M.; El-Molla, S.A. Textile dye removal from aqueous solutions using cheap MgO nanomaterials: Adsorption kinetics, isotherm studies and thermodynamics. Adv. Powder Technol. 2016, 27, 223-231.

Reviewer 3 Report

Comments and Suggestions for Authors

Review

Journal: Nanomaterials

Manuscript ID: nanomaterials-2727345-v2

Title: A novel nanoporous adsorbent for pesticides obtained from biogenic calcium carbonate, derived from waste crab shells

 The manuscript has certainly been improved and some of my comments were implemented; however, my most important concern was not addressed. What is more, some of the new results that were added to the revised paper are questionable at best and unacceptable at worst. Last, the manuscript is still full of English-related problems that, contrary to what is stated, have not been solved. Due to these concerns and the still-present obvious shortcomings, I cannot recommend the publication of this manuscript in its present form in a Q1 journal.

Scientific comments:

Comment #1: We will consider all these observations for the future elaborated studies that we will perform on this material.

Answer #1: Unfortunately, I cannot accept this answer. In my comment, I explicitly highlighted that there are questionable results presented in this manuscript, which must be addressed. The scientific literature should be considered and the dependence of each one of the investigated parameters on the adsorption properties should be discussed. A plausible explanation should be provided as to why the adsorption capacity changed so erratically, sometimes as a result of very small changes.

Comment #2: Added. (Error bars)

Answer #2: Please provide information in the experimental section regarding how many times the experiments were repeated and regarding the calculation of error bars. The inclusion of error bars confirmed the reliability of the results presented, which is a positive outcome; however, they make the results even more intriguing and the need for adding an explanation even more justified.

Comment #3: Added. (Isotherm model calculations)

Answer #3: It is stated that the Langmuir model fitted the adsorption data well (twice, in Lines 618 and 626) based on an R2 value of 0.714. In such an analytical study, this is an erroneous statement. In a study that investigates biological systems, an R2 value of 0.714 can indeed be considered a good fit, but not in this one. In an analytical study such as this one, where the parameters can be controlled well, only an R2 value above 0.9 can be considered a good fit. There are other adsorption models that should be considered with proper literature background.

Comments on the Quality of English Language

English- and writing-related comments:

The manuscript is still full of English-related problems. As a proofreader myself, I will give a demonstration based on only the abstract and the beginning of the introduction section.

1)      Line 14: a “the” is missing between “through” and “thermal” (“the” should be used for post-modifiers with non-partitive “of”)

2)     Lines 17-18: subject problem, the word “respectively” should be used instead: “BET surface area, BET pore volume, and pore diameter were 250 m2 g-1, 0.4 cm3 g-1, and <70 nm, respectively.”

3)     Line 19: “to be” is missing between “determined” and “around”

4)     Line 22: using “Under optimum conditions” would be better

5)     Lines 22-24: the sentence should be rephrased so that the sentence does not end with the predicate: “The adsorption capacity of BC1 was 17.8 mg g-1 under optimum conditions (i.e., …)”

6)     Lines 24-25: the sentence should be rephrased: “The equilibrium data obtained from the adsorption experiments fitted the Langmuir isotherm model well.”

7)     Lines 25-27: the sentence contains too many prepositions and is difficult to read, so it should be rephrased: “We developed an effective nanoporous adsorbent for recycling crab shells, which can be applied on site with minimal laboratory infrastructure according to local needs.”

8)     Line 33: “such” is missing between “application” and “as”

9)     Line 37: the “the” between “note” and “two” should be deleted

10) Line 40: “other” should be used instead of “another”

11) Line 42: “a” should be used instead of “as”

12) Line 45: a noun is missing after “and other similar,”

Author Response

Dear reviewer,

Thank you for your comments!

You can find our answers below and we hope that our paper is in accordance with the requirements.

Best regards,

Loredana Soran

Review

Journal: Nanomaterials

Manuscript ID: nanomaterials-2727345-v2

Title: A novel nanoporous adsorbent for pesticides obtained from biogenic calcium carbonate, derived from waste crab shells

The manuscript has certainly been improved and some of my comments were implemented; however, my most important concern was not addressed. What is more, some of the new results that were added to the revised paper are questionable at best and unacceptable at worst. Last, the manuscript is still full of English-related problems that, contrary to what is stated, have not been solved. Due to these concerns and the still-present obvious shortcomings, I cannot recommend the publication of this manuscript in its present form in a Q1 journal.

Scientific comments:

Comment #1: We will consider all these observations for the future elaborated studies that we will perform on this material.

Answer #1: Unfortunately, I cannot accept this answer. In my comment, I explicitly highlighted that there are questionable results presented in this manuscript, which must be addressed. The scientific literature should be considered and the dependence of each one of the investigated parameters on the adsorption properties should be discussed. A plausible explanation should be provided as to why the adsorption capacity changed so erratically, sometimes as a result of very small changes.

Answer:  Infrared adsorption studies have confirmed that carbonic acid has the main influence upon the surface chemistry of calcium carbonate in normal atmospheric conditions. This is also true for aqueous solutions, albeit the carbonic acid here rapidly dissolves into CO2 and H2O [39].

The solubility of calcium carbonate increases as the pH becomes increasingly acidic[40,41]. The pH = 8, at which maximum acetamiprid removal rate is observed, is below the point of zero charge (pHpzc = 9.14), which makes its surface positively charged and hence favorable for acetamiprid adsorption from electrochemical point of view. The solubility of calcium carbonate (and BC1 itself) increases as the pH becomes increasingly acidic. Thus, pH of 8 is still above the pH value where notable dissolution of CaCO3 occurs, which accounts for lower and erratic removal rate. In fact, below this pH, the calcite reacts with HCl from solution, a part of this being transformed in CaCl2, a soluble salt. Due to this fact, the adsorption can varies and the equilibrium is unstable at pH less than 8. On the other hand, to acidic pH, the acetamiprid molecule will be protonated and the interactions between CaCO3 and acetamiprid are modified. The variation of removal degree is due to cycles of adsorption – desorption

The solubility of both CO2 and CaCO3 in water decreases with solution temperature increasing [40, 41]. This indirectly influences the carbonic acid formation and dissolution. The range between 25 and 35 °C may be an anomalous interval where the carbonate chemistry is strongly favorable for acetamiprid adsorption. However, the current study cannot provide a definite answer for the underlying process.

The observations regarding removal rates at different contact time could be explained through the fact that due to the complex motions of the molecules within the solution, both adsorption and desorption processes occur. Increasing removal rates are observed at contact times until the first 15 minutes due to great availability of surface area for adsorption. Removal rates seem to decrease at contact times from 15 to 30 minutes due to the higher desorption rate resulting from reaching and possibly breaching of the BC1 capacity. At longer contact times, a balance between the two processes seems to be reached, whereby a similar amount of acetamiprid is adsorbed at any given moment.

The effects of quantity of BC1 and the initial pollutant dose can also be explained through the availability of adsorption surface area on BC1.

References

  1. Al-Hosney, H.A.; Grassian, V.H. Carbonic Acid: An Important Intermediate in the Surface Chemistry of Calcium Carbonate. J. Am. Chem. Soc. 2004, 126, 8068-8069.
  2. Coto, B.; Martos, C.; Pena, J.L.; Rodriguez, R.; Pastor, G. Effects in the solubility of CaCO3: Experimental study and model description. Fluid Phase Equilib. 2012, 324, 1-7.
  3. Phipps, J.; Lorusso, M. Dissolution Behaviour of Calcium Carbonate in Mildly Acidic Conditions. In: The science of papermaking (Baker C.F., ed.). Trans. of the XIIth Fund. Res. Symp. Oxford, 2001, pp 415–427. DOI: 10.15376/frc.2001.1.415.

These explanations were introduced in the “Results” section.

Comment #2: Added. (Error bars)

Answer #2: Please provide information in the experimental section regarding how many times the experiments were repeated and regarding the calculation of error bars. The inclusion of error bars confirmed the reliability of the results presented, which is a positive outcome; however, they make the results even more intriguing and the need for adding an explanation even more justified.

Answer: The following sentence was added in the figure 5 caption: “Each data point is the mean ± the standard error of the mean of three independent experiments.”

Comment #3: Added. (Isotherm model calculations)

Answer #3: It is stated that the Langmuir model fitted the adsorption data well (twice, in Lines 618 and 626) based on an R2 value of 0.714. In such an analytical study, this is an erroneous statement. In a study that investigates biological systems, an R2 value of 0.714 can indeed be considered a good fit, but not in this one. In an analytical study such as this one, where the parameters can be controlled well, only an R2 value above 0.9 can be considered a good fit. There are other adsorption models that should be considered with proper literature background.

Answer: Two intermediate points were added and the value of R2 increased to 0.9123.

English- and writing-related comments:

The manuscript is still full of English-related problems. As a proofreader myself, I will give a demonstration based on only the abstract and the beginning of the introduction section.

  • Line 14: a “the” is missing between “through” and “thermal” (“the” should be used for post-modifiers with non-partitive “of”)

Answer: Corrected

  • Lines 17-18: subject problem, the word “respectively” should be used instead: “BET surface area, BET pore volume, and pore diameter were 250 m2 g-1, 0.4 cm3 g-1, and <70 nm, respectively.”

Answer: Corrected

  • Line 19: “to be” is missing between “determined” and “around”

Answer: Corrected

  • Line 22: using “Under optimum conditions” would be better

Answer: Corrected

  • Lines 22-24: the sentence should be rephrased so that the sentence does not end with the predicate: “The adsorption capacity of BC1 was 17.8 mg g-1 under optimum conditions (i.e., …)”

Answer: Corrected

  • Lines 24-25: the sentence should be rephrased: “The equilibrium data obtained from the adsorption experiments fitted the Langmuir isotherm model well.”

Answer: Corrected

  • Lines 25-27: the sentence contains too many prepositions and is difficult to read, so it should be rephrased: “We developed an effective nanoporous adsorbent for recycling crab shells, which can be applied on site with minimal laboratory infrastructure according to local needs.”

Answer: Corrected

  • Line 33: “such” is missing between “application” and “as”

Answer: Corrected

  • Line 37: the “the” between “note” and “two” should be deleted

Answer: Corrected

  • Line 40: “other” should be used instead of “another”

Answer: Corrected

  • Line 42: “a” should be used instead of “as”

Answer: Corrected

  • Line 45: a noun is missing after “and other similar,”

Answer: Corrected

Answer: The English was revised.

Round 3

Reviewer 3 Report

Comments and Suggestions for Authors

The authors carried out the changes I requested acceptably. The manuscript can be published.

Comments on the Quality of English Language

There are still many mistakes left, but I reckon they will be corrected during proofreading and/or by the MDPI proofreaders.